# StateEnsemble: Neural State Chain-based Dynamic Model for Complex Motor Sequence Decoding

## Abstract

Brain-computer interfaces (BCIs) have paved the way for motor function rehabilitation and reconstruction. However, accurate movement decoding is still a challenging problem, especially for complex movements. Recent studies discovered that motor sequences, particularly complex ones, are encoded through a chain of neural states, each corresponding to a movement fragment. While this neural basis could facilitate more accurate neural decoding for complex movements, existing neural decoders fall short in modeling state-level sequential information. Here, we propose a neural state chain-based dynamic model (StateEnsemble), which explicitly models the neural state transition process to perform state-dependent neural decoding. We evaluated the proposed approach with intracortical neural signals recorded from the human motor cortex during handwriting. Experimental results demonstrated that our approach can effectively capture the underlying neural state transition patterns during handwriting, and achieve significant improvements in decoding performance. The proposed StateEnsemble approach can be beneficial for diverse neural decoding tasks and facilitate high-performance BCIs.

## 1 Introduction

Advances in brain-computer interfaces (BCIs) have paved a promising way for rehabilitation and reconstruction of motor function (Lorach et al., 2023; Moritz et al., 2024), and enabling control of prosthetic devices or computer cursors through neural activity (Chaudhary et al., 2016). Neural decoders, which translate neural signals into motor parameters, play a key role in BCIs (Taylor et al., 2002; Hochberg et al., 2006a; Shenoy & Carmena, 2014; Gilja et al., 2015; Schwemmer et al., 2018).

However, constructing a robust and accurate neural decoder is still a challenging problem for BCIs, and one underlying reason is the nonstationary property of neural signals (Kim et al., 2006a; Sanes & Donoghue, 2000; Churchland & Shenoy, 2007; Sussillo et al., 2016). Namely, neural tuning could change over time, which leads to unstable decoding performance. To obtain stable neural decoding against the neural nonstationarity, some studies proposed to periodically recalibrate the neural decoders or manually adjust them when performance degrades (Hochberg et al., 2012; Gilja et al., 2015; Shanechi et al., 2016; Brandman et al., 2018). Despite its effectiveness, the recalibration process commonly disrupts user experience (Murphy et al., 2016), and is mostly insufficient for short-term instability, such as those that occur within single trials (Khademi et al., 2023). Other studies aim to construct dynamic models to track the nonstationary changes in neural signals (Eden et al., 2004; Wang & Principe, 2008; Qi et al., 2019; Zhu et al., 2022; Qi et al., 2022). These methods avoid the recalibration process and are potentially more suitable for long-term use (Wang et al., 2015). However, since nonstationary changes in neural signals could stem from diverse origins such as noise (Collinger et al., 2013), unstable recording situations (Churchland & Shenoy, 2007; Sussillo et al., 2016), or task-related variables (Suway et al., 2018), the modeling of such complex neural nonstationarity is a challenging problem.

Recent studies demonstrated that a complex movement sequence contains a chain of neural states, each of which covers a small fragment of movement (Horrocks et al., 2024; Roth & Ding, 2024; Simpson, 2024; Qi et al., 2025), which provides a novel way to understand and model the nonstationary changes in neural signals. However, existing neural decoders fall short in modeling the sequential relationship

Figure 1: Neural state chain-based dynamic model (StateEnsemble) framework.

between neural states. Although some models considered the neural states in decoding (Qi et al., 2019; 2022), they ignored the state transition process, especially the state transition patterns that may exist in the state chain, leading to unstable decoding performance.

To address this problem, we propose a neural state chain-based dynamic model (StateEnsemble), which explicitly models the neural state transition process and performs state-dependent neural decoding. StateEnsemble contains an encoding phase that identifies the neural state chain and constructs the state transition graph, and a decoding phase that performs state-dependent decoding using the state transition graph as the prior. We evaluated the proposed approach using intracortical neural signals recorded from the human motor cortex during the task of writing Chinese characters. Experimental results showed that our method can effectively capture stable neural state transition patterns across repetitive writing trials. The proposed neural decoder outperforms baseline methods in decoding movement trajectories.

## 2 THE STATEENSEMBLE MODEL

### 2.1 CLASSIC STATE-SPACE MODEL

Classic state-space model consists of a state transition function $f(.)$ and a measurement function $h(.)$ as follows (Friedland, 2012):

$$\mathbf{x}_k = f(\mathbf{x}_{k-1}) + \mathbf{v}_{k-1}, \tag{1}$$

$$\mathbf{y}_k = h(\mathbf{x}_k) + \mathbf{w}_k. \tag{2}$$

Let $k$ denote the discrete time step, $\mathbf{x}_k \in \mathbb{R}^{d_x}$ the state of our interest, $\mathbf{y}_k \in \mathbb{R}^{d_y}$ the measurement or observation. Additionally, $\mathbf{v}_k$ and $\mathbf{w}_k$ represent i.i.d. state transition noise and measurement noise.

In the context of neural decoding, the state and observation represent the movement trajectory and the neural signals, respectively. A measurement function is referred to as a neural state. Given a sequence of neural signals $\mathbf{y}_{0:k}$, the state-space model can recursively estimate the posterior probability density $p(\mathbf{x}_k|\mathbf{y}_{0:k})$.

## 2.2 STATE-SPACE MODEL WITH A NEURAL STATE CHAIN

Recent neuroscience studies showed that a motion sequence consists of a chain of neural states (Horrocks et al., 2024; Roth & Ding, 2024; Simpson, 2024; Qi et al., 2025). Classic state-space models assume that the measurement function $h(.)$ does not change over time, thus showing unstable performance given temporal variability in neural signals caused by neural state switching (Kim et al., 2006b). The existing dynamic models mostly did not explicitly consider the state transition process, especially the state transition patterns that may exist in the state chain, which can lead to unstable and inaccurate decoding performance.

Here we aim to incorporate the neural state transition process into the state-space models. *Note that the 'state' can represent both the state $\mathbf{x}$ in the state transition function and the neural state $h(\cdot)$. Thus, we refer to the state in the state transition function by 'movement trajectory' and the neural state by 'state' in the motor decoding scenario.*

We hypothesize that the neural state sequence contains a certain state transition pattern and propose to explicitly model the transition pattern with a graph. Specifically, we formulate the neural state chain in our model (StateEnsemble) as follows:

$$\mathbf{x}_k = f(\mathbf{x}_{k-1}) + \mathbf{v}_{k-1}, \tag{3}$$

$$\mathbf{y}_k = h_k(\mathbf{x}_k) + \mathbf{w}_k, \tag{4}$$

$$h_k = g(h_{k-1}). \tag{5}$$

Let $g(.)$ denote the neural state transition graph embedded in the state chain, where $g(h_{k-1}) = p(h_k|h_{k-1})$, with $p(h_k|h_{k-1})$ representing the transition probability from $h_{k-1}$ to $h_k$. Let the set of $M$ neural states be $\mathbb{H} = \{h^1, h^2, \ldots, h^M\}$, with $h_k \in \mathbb{H}$ being the neural state at time step $k$. The state transition graph $g(.)$ is represented as an $M \times M$ matrix. In this matrix, the element in the $i$-th row and the $j$-th column is $p(h^j|h^i)$, and $p(h^i|h^i) > p(h^j|h^i)$ for $i \neq j$, indicating that self-transitions occur most frequently.

To estimate the movement trajectory given neural signals with the StateEnsemble model, two materials are required: the neural-to-motor mapping function of each state and the state transition graph, which we should estimate in the training process. In the predicting process, the StateEnsemble model receives incoming neural signals, identifies the current state, and switches to the proper state functions to predict the movements. Thus, there are two important questions to answer: 1) In the training process, how to identify the neural states and how to model the state transition graph; 2) In the prediction process, how to perform state-dependent decoding such that the decoder can adaptively switch to the proper model along with changes in states. The framework of StateEnsemble is illustrated in fig. 1.

## 2.3 NEURAL STATE IDENTIFICATION AND STATE TRANSITION GRAPH

In the training process, our first objective is to estimate the neural state chain given a sequence of neural signal-movement trajectory pairs of $o_{0:n}$, where $o_k := \{\mathbf{x}_k, \mathbf{y}_k\}$, and then construct the state transition graph based on the chain. We assume that the functional mapping between the neural signal and movement trajectory is stable within a state, while distinct among different states.

Here, we propose the **state identification algorithm**. Given the number of neural states $M$, it first initializes a set of functions with a randomly generated state transition graph, each defining a functional mapping between the neural signals and the movement trajectory. Then it applies the function and graph to each data pair of $o_{0:n}$, and each data will be assigned to a neural state by maximizing the posterior probability of the states, such that a neural state chain can be constructed. After that, the state transition graph can be obtained given the state chain, and the functional mappings of the states can be updated with the data pairs assigned to each state. This process is repeated iteratively until the state transition graph stabilizes.

1) Initialization:

To initialize the neural state chain and establish functional mappings between the movement trajectory and neural signals, i.e., $\mathbb{H} = \{h^1, h^2, \ldots, h^M\}$, we first randomly generate a state transition graph $g(.)$ with relatively high transition probabilities for self-transitions. Next, we randomly select a starting state and sample the neural state chain according to the transition graph. Based on the constructed state chain, we assign data pairs to neural states and fit the functional mappings for each state. The initial posterior probabilities of the neural states are set equal.

2) Data-function assignment:

We aim to assign the data pairs at each time step to one of the $M$ neural states. Specifically, at time $k$, we select the neural state $h^m$ that maximizes the posterior probability of $h_k$, i.e., $p(h_k|o_{0:k})$, where $h^m \in \mathbb{H} = \{h^1, h^2, \ldots, h^M\}$, given the observations $o_{0:k}$.

To recursively derive $p(h_k|o_{0:k})$ based on $p(h_{k-1}|o_{0:k-1})$ and the state transition graph $g(.)$, we apply the Bayesian update mechanism (Hoeting et al., 1999; Yuen & Kuok, 2011) and the Markov property (Norris, 1998). First, we predict the neural state at time $k$ as follows:

$$p(h_k|o_{0:k-1}) = \sum_{h_{k-1} \in \mathbb{H}} p(h_k|h_{k-1})p(h_{k-1}|o_{0:k-1}). \tag{6}$$

Next, applying Bayesian rule, the posterior probability of the neural state at time $k$ is:

$$p(h_k|o_{0:k}) = \frac{p(h_k|o_{0:k-1})p(o_k|o_{0:k-1}, h_k)}{\sum_{h_k \in \mathbb{H}} p(h_k|o_{0:k-1})p(o_k|o_{0:k-1}, h_k)}. \tag{7}$$

The term $p(o_k|o_{0:k-1}, h_k)$ can be derived using the first-order Markov property as follows:

$$\begin{aligned} p(o_k|o_{0:k-1}, h_k) &= p(\mathbf{x}_k, \mathbf{y}_k|o_{0:k-1}, h_k) \\ &= p(\mathbf{y}_k|\mathbf{x}_k, o_{0:k-1}, h_k)p(\mathbf{x}_k|o_{0:k-1}, h_k) \\ &= p(\mathbf{y}_k|\mathbf{x}_k, h_k)p(\mathbf{x}_k|\mathbf{x}_{k-1}), \end{aligned} \tag{8}$$

where $p(\mathbf{y}_k|\mathbf{x}_k, h_k)$ is the likelihood function associated with the neural state at time $k$, and $p(\mathbf{x}_k|\mathbf{x}_{k-1})$ is the prior probability of the movement at time $k$.

3) State transition graph updating:

Based on the neural state chain constructed in each iteration, we could update the state transition graph $g(.)$ statistically by counting the number of state transitions in the chain. Specifically, for each pair of states $h^i$ and $h^j$, we count how often the transition from $h^i$ to $h^j$, occurs in the neural state chain. This gives an empirical estimate of the transition probabilities $p(h^j|h^i)$, which are then normalized to ensure that the sum of transition probabilities to any given state equals 1. The updated state transition graph $g(.)$ is then used as the prior in the next iteration of the algorithm.

4) State function updating:

We update the functional mappings for the $M$ neural states based on the new assignment of data pairs obtained in the second step. The function fitting process can utilize various methods, including least squares estimation, nonlinear fitting, and neural networks.

The state identification algorithm is given in algorithm S1.

### 2.4 STATE-DEPENDENT NEURAL DECODING USING THE STATE TRANSITION GRAPH AS A PRIOR

In an online decoding process, the objective is to recursively estimate the posterior probability of the movement trajectory at time $k$, $\mathbf{x}_k$, given a sequence of neural signals $\mathbf{y}_{0:k}$. The DyEnsemble approach (Qi et al., 2019) can perform state-dependent neural decoding in a Bayesian filter framework, where it first estimates the posterior probability of states at each time step, and then adaptively weighs and ensembles a decoding model. However, the DyEnsemble model does not model the neural state transition process. Thus, in this study, we propose to solve the StateEnsemble model by incorporating the state transition graph with DyEnsemble, serving as a prior in the state estimation process.

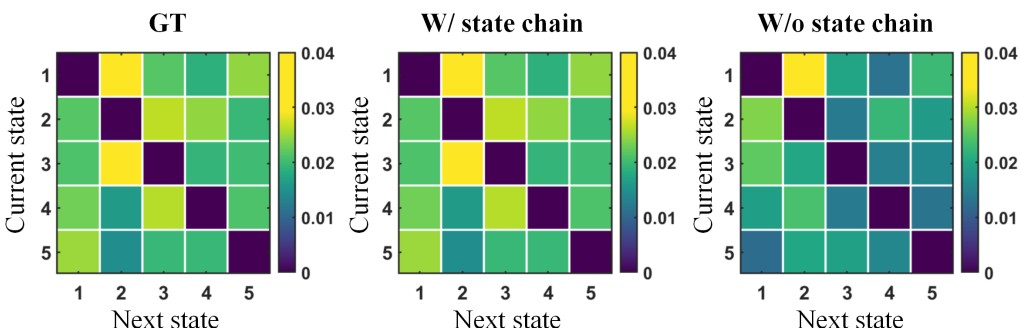

Figure 2: The state transition graph of 5 neural states is presented in matrix form, where each row corresponds to the current state, and each column corresponds to the next state. The element in the $i$-th row and $j$-th column represents the transition probability from state $i$ to state $j$.

Specifically, a Bayesian updating mechanism is used to dynamically switch among these neural states in a data-driven manner. Given the observation sequence $\mathbf{y}_{0:k}$, the posterior of the movement trajectory at time $k$ is given by:

$$p(\mathbf{x}_k|\mathbf{y}_{0:k}) = \sum_{h_k \in \mathbb{H}} p(\mathbf{x}_k|\mathbf{y}_{0:k}, h_k)p(h_k|\mathbf{y}_{0:k}), \tag{9}$$

where $p(\mathbf{x}_k|\mathbf{y}_{0:k}, h_k)$ is the posterior probability of the movement corresponding to the neural state $h_k$, and $p(h_k|\mathbf{y}_{0:k})$ is the posterior probability of the neural state $h_k$. Given the neural signals, we should estimate both the movement $\mathbf{x}_k$ and the neural state $h_k$.

First, we estimate the posterior probability of the neural state. Using the state transition graph as the prior, the neural state at time $k$ can be predicted as follows:

$$p(h_k|\mathbf{y}_{0:k-1}) = \sum_{h_{k-1} \in \mathbb{H}} p(h_k|h_{k-1})p(h_{k-1}|\mathbf{y}_{0:k-1}). \tag{10}$$

Employing Bayes' rule, the posterior probability of the neural state at time $k$ can be obtained by:

$$p(h_k|\mathbf{y}_{0:k}) = \frac{p(h_k|\mathbf{y}_{0:k-1})p(\mathbf{y}_k|\mathbf{y}_{0:k-1}, h_k)}{\sum_{h_k \in \mathbb{H}} p(h_k|\mathbf{y}_{0:k-1})p(\mathbf{y}_k|\mathbf{y}_{0:k-1}, h_k)}. \tag{11}$$

The term $p(\mathbf{y}_k|\mathbf{y}_{0:k-1}, h_k)$ represents the marginal likelihood of the neural state at time $k$, which is defined as:

$$p(\mathbf{y}_k|\mathbf{y}_{0:k-1}, h_k) = \int p(\mathbf{y}_k|\mathbf{x}_k, h_k)p(\mathbf{x}_k|\mathbf{y}_{0:k-1}, h_k)d\mathbf{x}_k, \tag{12}$$

where $p(\mathbf{y}_k|\mathbf{x}_k, h_k)$ is the likelihood function associated with the neural state at time $k$, and the movement at time $k$, $p(\mathbf{x}_k|\mathbf{y}_{0:k-1}, h_k)$ can be predicted as:

$$p(\mathbf{x}_k|\mathbf{y}_{0:k-1}, h_k) \approx \int p(\mathbf{x}_k|\mathbf{x}_{k-1})p(\mathbf{x}_{k-1}|\mathbf{y}_{0:k-1})d\mathbf{x}_{k-1}, \tag{13}$$

where $p(\mathbf{x}_k|\mathbf{x}_{k-1})$ is the prior probability of the movement at time $k$ and $p(\mathbf{x}_{k-1}|\mathbf{y}_{0:k-1})$ is the posterior probability of the movement at time $k-1$.

The particle filtering (PF) was employed to approximate the posterior distribution of movement corresponding to the neural state $h_k$, $p(\mathbf{x}_k|\mathbf{y}_{0:k}, h_k)$, with a weighted particle set (Arulampalam et al., 2002). The posterior $p(\mathbf{x}_k|\mathbf{y}_{0:k})$ can then be recursively estimated with particles.

## 3 EXPERIMENTS WITH SIMULATION DATA

We firstly evaluated the StateEnsemble model on simulation data to examine two key aspects: (1) in the training process, to assess whether the state identification algorithm can effectively capture the

state transition graph embedded in the neural state chain and fit the functional mappings of different neural states better than methods without the graph as a prior; and (2) in the decoding phase, to evaluate whether the state-dependent decoding algorithm that uses the state transition graph as a prior achieves superior performance compared to methods that do not use it. To achieve these objectives, we design comparison experiments that contrast the training and prediction performance of the "with state chain" method (using the state transition graph) and the "without state chain" method (ignoring the graph).

**With state chain:** In both the training and prediction process, the method uses the state transition graph as a prior to guide the state transition process, following the details described in section 2.

**Without state chain:** In both the training and prediction process, the method does not utilize the prior guidance provided by the state transition graph. Specifically: In the training process, during the data-function assignment step, data pairs are assigned by minimizing the predicted error across all neural states, as follows:

$$h_k = \arg \min_{h_k \in \mathbb{H}} ||\mathbf{y}_k - h_k(\mathbf{x}_k)||_2. \tag{14}$$

In the prediction process, the method follows the DyEnsemble approach, which does not explicitly model the neural state chain.

### 3.1 SIMULATION OF NEURAL SIGNALS WITH A NEURAL STATE CHAIN

For the simulation, we generate the movement trajectory $\mathbf{x}_{0:n}$ using the motion sequence from a monkey self-paced reaching task (O'Doherty et al., 2017). Additionally, we generate the neural signal sequence $\mathbf{y}_{0:n}$, the state transition graph $g(.)$, and the functional mappings of the neural states $\mathbb{H} = \{h^1, h^2, \ldots, h^M\}$. To simplify the analysis, we set $M = 5$ neural states, with the movement dimension $d_x = 2$ and the neural signal dimension $d_y = 70$ (corresponding to 70 neurons). The mappings of different neural states are modeled as linear functions with biases. The data size $n$ is set to 5 times the number of parameters in the mapping for a single neural state to ensure proper model fitting. Further details on the data generation process are provided in section B.1.

### 3.2 PERFORMANCE OF STATE TRANSITION GRAPH LEARNING

The comparison results for the state transition graph $g(.)$ among the ground truth (GT), "with state chain" and "without state chain" are presented in two forms: the matrix form in fig. 2 and the graph form in fig. S1. To improve readability, diagonal elements—representing self-transition probabilities—are omitted, as they are generally higher than inter-state transition probabilities. However, their omission does not affect the comparative analysis of inter-state transitions. Both figures suggest that the transition patterns in the "with state chain" method exhibit greater similarity to the ground truth than those in the "without state chain" method. The transition probabilities in fig. S1 demonstrate that "with state chain" accurately reproduces the ground truth state transition graph, while "without state chain" results in a less precise approximation. This highlights the effectiveness of incorporating a state transition graph as a prior, allowing the algorithm to learn the inherent structure of neural state transitions.

### 3.3 PERFORMANCE OF NEURAL STATE FUNCTIONAL MAPPING ESTIMATION

We computed the mean squared error (MSE) between the estimated parameters of the functional mappings and the ground truth for different neural states using the "with state chain" and "without state chain" methods, as shown in fig. S2(a). On average, the MSE for the "with state chain" method is reduced by 41.97% ($p < 0.001$) compared to the "without state chain" method. Additionally, the estimates obtained with the "with state chain" method exhibit a lower standard deviation, indicating improved consistency and stability in parameter estimation.

To visually assess estimation accuracy, we plot the two-dimensional encoding parameters of a single neuron across five neural states in fig. S2(b). The visualization shows that the points corresponding to "with state chain" are consistently closer to the ground truth across all neural states, indicating that the prior constraints contribute to more accurate parameter estimates. A more detailed analysis is provided in section B.2.

## 3.4 Performance of neural decoding

The decoding task aims to estimate the movement trajectory $\mathbf{x}_{0:n}$ given the neural signal sequence $\mathbf{y}_{0:n}$, utilizing the training results. Both the "with state chain" and "without state chain" methods use their respective estimates of the functional mappings of neural states, $\mathbb{H} = \{h^1, h^2, \ldots, h^M\}$; however, the "with state chain" method incorporates the guidance from the state transition graph $g(.)$ as a prior, while "without state chain" does not leverage this prior information.

The comparison of decoding performance, measured by the MSE of the movement trajectory, is shown in fig. S3. The results show that the MSE of "with state chain" is significantly lower ($p < 0.001$) and more stable than that of "without state chain." This highlights the advantage of using the state transition graph embedded in the neural state chain, which improves both the accuracy and stability of the decoding process. We also evaluated the performance of our model under different experimental settings. The detailed analysis is in section B.3.

## 4 Experiments with neural signals

### 4.1 Neural signals of handwriting task

We applied our approach to a dataset involving imagined handwriting, a complex motor control task. Neural signals were recorded using two 96-channel Utah microelectrode arrays implanted in the left-hand knob area of the precentral gyrus. During the experiments, the participant was instructed to imagine writing Chinese characters stroke by stroke while following a virtual hand displayed in a video. Each experimental session consisted of 30 unique characters, with each character repeated three times in a randomized order, resulting in a total of 90 trials per session. Data from 18 experimental sessions were collected and analyzed. Specifically, we analyzed the sorted single-unit activities alongside the corresponding target velocity data.

### 4.2 Analysis of the state transition graph during handwriting

In the encoding experiments, we investigate whether a state transition graph is embedded in the neural state chain of handwriting tasks. For each session, the data is randomly divided into two groups, each consisting of motion and neural data for three repetitions of 15 characters, with no overlap of characters between the two groups. If the state transition graphs learned from these two groups are similar, it would suggest the presence of transition patterns in the neural state transition processes during handwriting tasks.

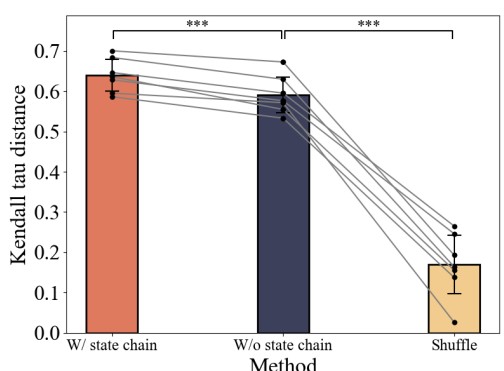

Figure 3: The Kendall tau distance of state transition graphs of two groups is shown in this graph. The points of the same session are connected by gray lines.

Using the "with state chain" and "without state chain" methods described in section 3, we explicitly and implicitly obtain the neural state chain and construct the corresponding state transition graph for each group. To quantify the similarity between the learned graphs, we calculate the Kendall tau distance by ranking transition probabilities and comparing the relative ordering of state transitions. We also generate random shuffle graphs. These graphs are created by shuffling the order of states in the neural state chain, thus eliminating any inherent transition patterns in the chain. These shuffle graphs are compared with each other to establish a baseline of similarity between graphs of two groups, which helps us understand how different the learned graphs are from random, non-structured patterns.

The results in fig. 3 show that the Kendall tau distance for shuffle graphs is significantly lower than for both the "with state chain" and "without state chain" methods ($p < 0.001$), indicating that a distinct transition pattern exists during neural state transitions while handwriting. Moreover, the "with state

chain" method learns more similar state transition graphs across two groups than the "without state chain" method. This suggests that the state transition graph prior enhances the stability and accuracy of the learned graph.

We also visualize the state transition graphs learned by the "with state chain" method, as shown in fig. 4. In this visualization, we observe distinct transition structures, such as the "5 points of the dice" pattern on the left side of the graph (highlighted in the red box with solid lines). Additionally, there are other similar transition structures, with varying relative color depths (highlighted in the red box with dotted lines), rather than absolute color depth. The differences in absolute color depth, which represents the transition probability, may be influenced by the size of the samples. These results demonstrate that a structured transition graph is embedded in the neural state chain of handwriting tasks and that our method is capable of effectively capturing and learning these transition patterns by leveraging the state transition graph as a prior.

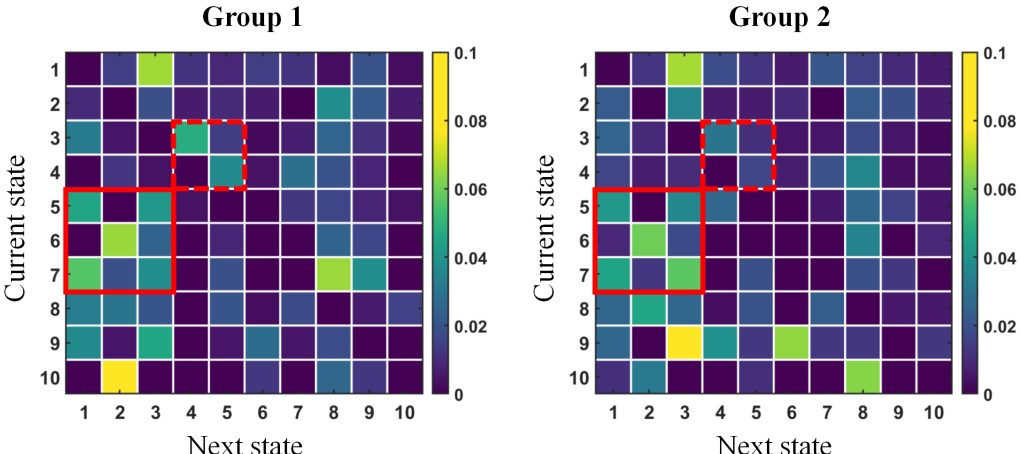

Figure 4: The state transition graphs of two groups learned by "with state chain" method are shown in this graph, where each row corresponds to the current state, and each column corresponds to the next state. **left)** group 1; **right)** group 2. Red boxes with solid lines indicate similar transition structures based on absolute depth, whereas red boxes with dotted lines indicate similar transition structures based on relative depth.

| Method | Session1 | Session7 | Session8 | Session10 | Session14 | Average |
|--------|----------|----------|----------|-----------|-----------|---------|
| KF | 0.1434 | 0.2875 | 0.1964 | 0.0918 | 0.2331 | $0.1008 \pm 0.0893$ |
| WF | 0.3364 | 0.1398 | 0.1272 | 0.3378 | 0.4241 | $0.2624 \pm 0.0975$ |
| WCF | 0.3474 | 0.1405 | 0.1292 | 0.3501 | 0.4328 | $0.2729 \pm 0.1004$ |
| RNN | 0.4090 | 0.4966 | 0.3985 | 0.4407 | 0.4950 | $0.3493 \pm 0.1023$ |
| LSTM | 0.4079 | 0.5419 | 0.4135 | 0.4695 | 0.5419 | $0.3732 \pm 0.1020$ |
| BIOT | 0.2715 | 0.4280 | 0.3683 | 0.3795 | 0.4956 | $0.2943 \pm 0.1134$ |
| BrainBERT | 0.4075 | 0.5052 | 0.4574 | 0.3888 | 0.5201 | $0.3759 \pm 0.0871$ |
| NDT2 | 0.4026 | 0.5088 | 0.4470 | 0.4168 | 0.5167 | $0.3786 \pm 0.0863$ |
| Ours w/o SC | 0.4049 | 0.5038 | 0.4406 | 0.4224 | 0.5220 | $0.3623 \pm 0.0948$ |
| **Ours** | **0.4151** | **0.5441** | **0.4689** | **0.4709** | **0.5436** | $\mathbf{0.3887 \pm 0.0978}$ |

Table 1: $R^2$ values for selected 5 sessions and the average across all 18 sessions between the true and estimated writing movement trajectories. The table compares the decoding performance of KF, WF, WCF, RNN, LSTM, BIOT, BrainBERT, NDT2, StateEnsemble without the state chain (Ours w/o SC), and StateEnsemble (Ours).

### 4.3 Performance of handwriting trajectory decoding

In the decoding phase, we compare the StateEnsemble and StateEnsemble without state chain methods against several established neural decoding approaches. The Kalman Filter (KF) (Stavisky et al., 2015) is a classical state-space model, while the Wiener Filter (WF) (Hochberg et al., 2006b) assumes a linear relationship between neural signals and motor parameters; the Wiener Cascade Filter (WCF) (Flint et al., 2013) extends WF by fitting a third-order polynomial between the WF output and the movement trajectory. Recurrent Neural Networks (RNNs) (Glaser et al., 2020) and Long Short-Term Memory Networks (LSTMs) (Costello et al., 2024) are widely used deep learning models for sequential data, with LSTMs representing a state-of-the-art approach for neural decoding. Transformer-based neural decoders, including BIOT (Yang et al., 2023), BrainBERT (Wang et al., 2023), and NDT2 (Ye et al., 2023), have also recently achieved state-of-the-art performance in neural decoding tasks. The implementation details are provided in section C.2.

Each session consists of a dataset of 30 Chinese characters, which is partitioned into two groups: 20 characters for training and 10 for testing. The training and testing sets contain no overlapping characters. This partitioning is repeated three times per session, each time using a different test set. To quantitatively evaluate decoding performance, we computed the $R^2$ values between the true and estimated writing movement trajectories. Results for five selected sessions and the average across all 18 sessions are shown in table 1, while the session-wise results for all 18 sessions are presented in table S1. The wide error bars are primarily due to the inherent variability in decoding performance across different sessions.

Classical decoding methods, such as KF and WF, are constrained by their linear assumptions, which limit their representational capacity and result in suboptimal performance. Although WCF introduces nonlinearity, its expressive power remains limited, yielding only marginal improvements over WF. In contrast, our method achieves a 42.43% increase in $R^2$ ($p < 0.01$) compared to WCF. Deep learning-based approaches, including RNNs, LSTMs, and Transformer-based decoders, further enhance decoding performance. Among these, LSTM and Transformer models exhibit superior capabilities, attributed to LSTM's gating mechanisms and Transformer's self-attention mechanism. Notably, the "without state chain" variant of our model (StateEnsemble without state chain) achieves performance comparable to LSTM and Transformer models. By contrast, the "with state chain" model (StateEnsemble) surpasses them, achieving a 4.15% increase ($p < 0.01$) relative to LSTM and a 2.66% increase over the Transformer-based decoder NDT2. These results highlight the effectiveness of incorporating state-dependent decoding for complex motor tasks such as handwriting. Comparing the "without state chain" and "with state chain" variants, incorporating the state transition prior yields a 7.29% increase ($p < 0.01$). By leveraging the constraints encoded in the state transition graph, the "with state chain" method promotes more coherent and stable transitions between neural states, thereby improving both the accuracy and stability of the decoded writing trajectories. These findings demonstrate that explicitly modeling neural state transitions can substantially enhance decoding performance in complex motor tasks.

We visualize the estimated writing movement trajectories reconstructed using the "with state chain" and "without state chain" methods in fig. S6. The detailed analysis of it is in section C.3. These results demonstrate that the state transition graph improves the stability and accuracy of decoding complex movement trajectories. It enables the decoding process to maintain the current neural state when necessary and promptly transition to the correct subsequent state. The proposed neural decoder outperforms baseline methods in decoding movement trajectories.

## 5 Conclusion

We propose a neural state chain-based dynamic model (StateEnsemble) that explicitly models the neural state transition process and performs state-dependent neural decoding. The proposed approach was evaluated using intracortical neural signals recorded from the human motor cortex during the writing of Chinese characters, a representative task that involves complex motor control. Experimental results demonstrated that StateEnsemble can effectively capture stable neural state transition patterns across repetitive writing trials. Moreover, the proposed neural decoder achieves superior performance in decoding movement trajectories, consistently outperforming baseline methods. These findings highlight the potential of StateEnsemble as a robust solution for motor sequence decoding, particularly in scenarios involving nonstationary neural signals and complex motor tasks.

## ETHICS STATEMENT

All clinical and experimental procedures in this study were reviewed and approved by the local Medical Ethics Committee and were formally registered. Informed consent was obtained verbally from the participant, along with consent from his family members, and was duly documented and signed by his legal representative.

The volunteer participant was a right-handed male, 75 years old at the time of data collection. He had sustained complete tetraplegia following a traumatic cervical spine injury at the C4 level due to a car accident approximately two years prior to study enrollment. The participant retained the ability to move body parts above the neck and demonstrated normal linguistic competence and comprehension for all tasks.

Two 96-channel intracortical microelectrode arrays (4 mm × 4 mm Utah arrays, 1.5 mm length; Blackrock Microsystems) were implanted in the left motor cortex (MC), with one array positioned in the center of the hand knob area (array A) and the second array located medially approximately 1 cm apart (array B). Neural signals were recorded via the implanted arrays using the NeuroPort system (NSP, Blackrock Microsystems). The signals were amplified, digitized, and sampled at 30 kHz.

The experimental task involved attempted handwriting: the participant observed a virtual hand writing a target character and simultaneously attempted to reproduce the same movement as if the virtual hand were his own. The instructional videos presented each character stroke by stroke to guide the participant's attempted movement.

## REPRODUCIBILITY STATEMENT

All experimental settings and implementation details are provided to facilitate reproducibility. Specifically, the generation of simulation data is detailed in section B.1; the algorithms for encoding and decoding are described in section 2.3, section 2.4, and algorithm S1; the neural data preprocessing procedures are provided in section 4.2 and section 4.3; the implementation details of baseline models are presented in section C.2; and the computational resources and complexity analysis are outlined in section C.1.

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

# A ALGORITHM

In this section, we write the algorithm of state identification in algorithm S1.

---

**Algorithm S1** State Identification Algorithm

---

**Input:** $n$: data size, $o_{0:n} = \{\mathbf{x}_{0:n}, \mathbf{y}_{0:n}\}$: observation data pairs, $M$: the number of neural states, $\alpha$: threshold

**Init:** $g(.)$: the state transition graph, $C$: the neural state chain, $\mathbb{H} = \{h^1, h^2, \ldots, h^M\}$: functional mappings of neural states, $Post$: the initial posterior probabilities of neural states

**repeat**

    **Data-function assignment:**

    $Like = \text{Likelihood}(o_{0:n}, C, H)$

    $Prior = \text{Prior}(g, Post)$

    $Post = \text{Posterior}(Prior, Like)$

    $C = \text{Chain}(Post)$

    **State transition graph updating:**

    $[g, changes] = \text{Graph-Update}(g, C)$

    **State function updating:**

    $\mathbb{H} = \text{Model-Fitting}(o_{0:n}, C, M)$

**until** $changes < \alpha$

**Output:** $\mathbb{H} = \{h^1, h^2, \ldots, h^M\}$: functional mappings of neural states, $g(.)$: the state transition graph

---

# B EXPERIMENTS WITH SIMULATION DATA

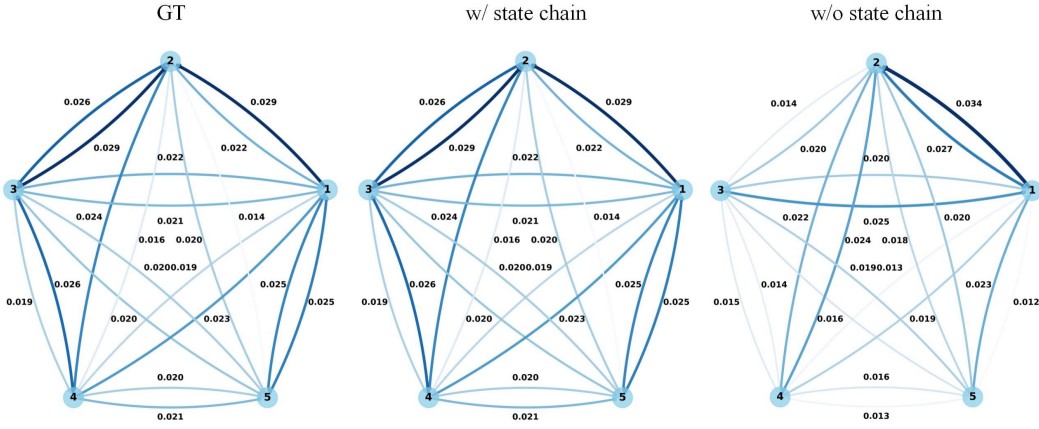

Figure S1: The state transition graph of 5 neural states is shown in graph form. Each node represents a neural state, and the weight of the directed edge from node $i$ to node $j$, labeled at the center of the edge, indicates the transition probability from state $i$ to state $j$.

## B.1 SIMULATION DATA GENERATION DETAILS

First, we use the motion sequence of a monkey self-paced reaching task for the movement trajectory. To construct a state transition graph, we initialize the diagonal elements of the state transition graph $g(.)$ to 0.9, ensuring that the neural state switches to itself most frequently. The transition probabilities to other neural states are sampled from a standard Gaussian distribution and normalized. Using this state transition graph, we then sample the neural state chain $h_{0:n}$ and regenerate the state transition graph $g(.)$ with the sampled sequence. Next, we generate $M$ linear functions with biases

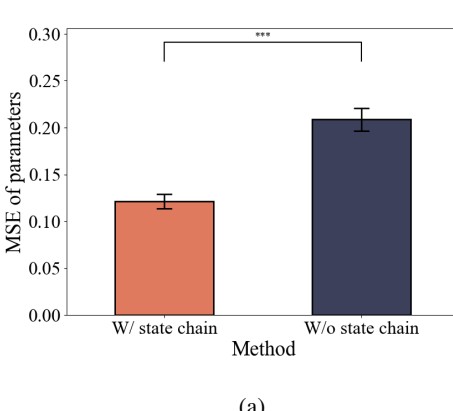 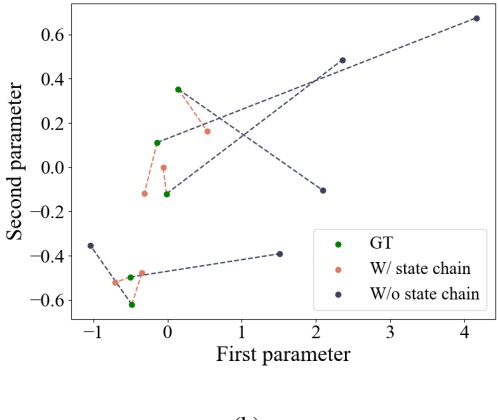

(a)                                                             (b)

Figure S2: (a) The MSE of the parameter estimates is shown in this graph. The training process was repeated 10 times with random initialization for both the "with state chain" and "without state chain" methods. (b) The parameters of 5 neural states for a neuron are visualized in this graph. Each point represents the parameters for a specific neural state of a neuron, where the horizontal coordinate corresponds to the first parameter and the vertical coordinate corresponds to the second parameter. Points belonging to the same neural state are connected by dotted lines. The points representing the ground truth are colored green, those for "with state chain" are colored orange, and those for "without state chain" are colored dark gray.

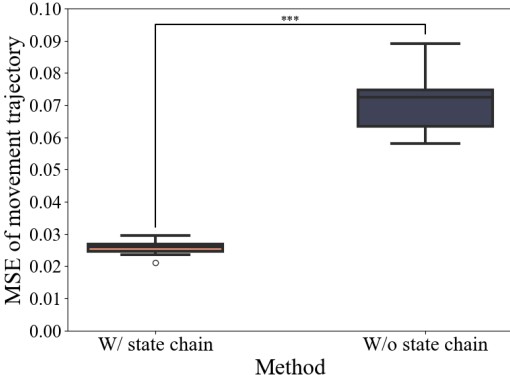

Figure S3: The MSE of the movement trajectory $x_{0:n}$ is shown in this graph. The decoding process was repeated 10 times with random initialization for both the "with state chain" and "without state chain" methods.

for $\mathbb{H} = \{h^1, h^2, \ldots, h^M\}$, parameters of which sampled from a standard Gaussian distribution. Finally, the neural signal sequence $y_{0:n}$ is generated based on $x_{0:n}$ using eq. (4), with Gaussian white noise $w_k$ added to the measurements.

### B.2 PERFORMANCE OF NEURAL STATE FUNCTIONAL MAPPING ESTIMATION

We computed the mean squared error (MSE) between the estimated parameters of the functional mappings and the ground truth for different neural states using the "with state chain" and "without state chain" methods, as shown in fig. S2(a). On average, the MSE for the "with state chain" method is reduced by 41.97% ($p < 0.001$) compared to the "without state chain" method. Additionally, the estimates obtained with the "with state chain" method exhibit a lower standard deviation, indicating improved consistency and stability in parameter estimation.

To visually assess estimation accuracy, we plot the two-dimensional encoding parameters of a single neuron across five neural states in fig. S2(b). Each point represents the parameters of a specific state, with colors distinguishing the ground truth, "with state chain," and "without state chain" estimates. Dotted lines connect points of the same state, where shorter lines indicate closer alignment with the ground truth. The visualization shows that the points for "with state chain" are consistently closer to the ground truth across all neural states, indicating that the prior constraints help achieve more accurate estimates. Additionally, the distances between the "with state chain" estimates and the ground truth are more stable than those for "without state chain", suggesting that incorporating the state transition graph results in more stable performance.

### B.3 ROBUSTNESS ANALYSIS WITH VARIOUS EXPERIMENTAL SETTINGS

To evaluate the performance of our model (StateEnsemble) under different experimental settings, we vary the number of neurons to $\{30, 40, 50, 60, 70, 80\}$, and the number of neural states to 5 and 10. The detailed analysis is in section B.3. For each setting, we select a specific combination of neuron count and neural state number, and perform the training and prediction processes 10 times, randomly initializing each trial. In the training process, we record the MSE of the parameters of the functional mappings of different neural states, as shown in fig. S4, while in the prediction process, we record the MSE of the movement trajectory, as shown in fig. S5. For each number of neural states, the training and prediction performance of the "with state chain" method consistently outperforms the "without state chain" method across all neuron settings, both in terms of accuracy and stability. Additionally, when using the "with state chain" method, we observe that the MSEs in both processes tend to decrease as the number of neurons increases, likely due to the additional information provided by more neurons. For a fixed number of neurons, the "with state chain" method consistently achieves better performance than the "without state chain" method, regardless of whether there are 5 or 10 neural states.

In conclusion, the state transition graph, as a prior, significantly improves both the accuracy and stability of the training and prediction processes across a wide range of experimental settings.

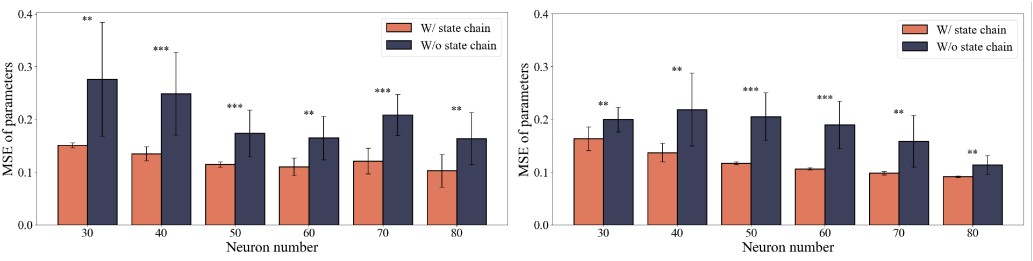

Figure S4: The statistical results of the MSE of the parameter estimates are shown in this graph. The number of neural states is set to 5 **(left)** and 10 **(right)**, and the number of neurons is set to $\{30, 40, 50, 60, 70, 80\}$. For each experimental setting, one specific combination of neuron number and neural state number is selected, and the training process is repeated 10 times.

## C EXPERIMENTS WITH NEURAL SIGNALS

### C.1 COMPUTE RESOURCES

The computational complexity of our method is similar to that of traditional particle filtering. Specifically, let $N$ denote the number of particles, $M$ the number of neural states, and $T$ the total number of timesteps, where $N \gg M$. In each timestep, the likelihood is calculated for each neural state and particle, resulting in a complexity of $O(NM)$. We then use the prior and likelihood to compute the posterior of the neural states and particles, which has a complexity of $O(M + M(N-1) + (M-1))$ for neural states and $O(NM + NM + N(M-1))$ for particles. Thus, the total time complexity per timestep is $O(NM)$, and for all timesteps, the overall complexity is $O(NMT)$. While we conducted only off-line experiments, based on the experimental results, it is feasible to achieve

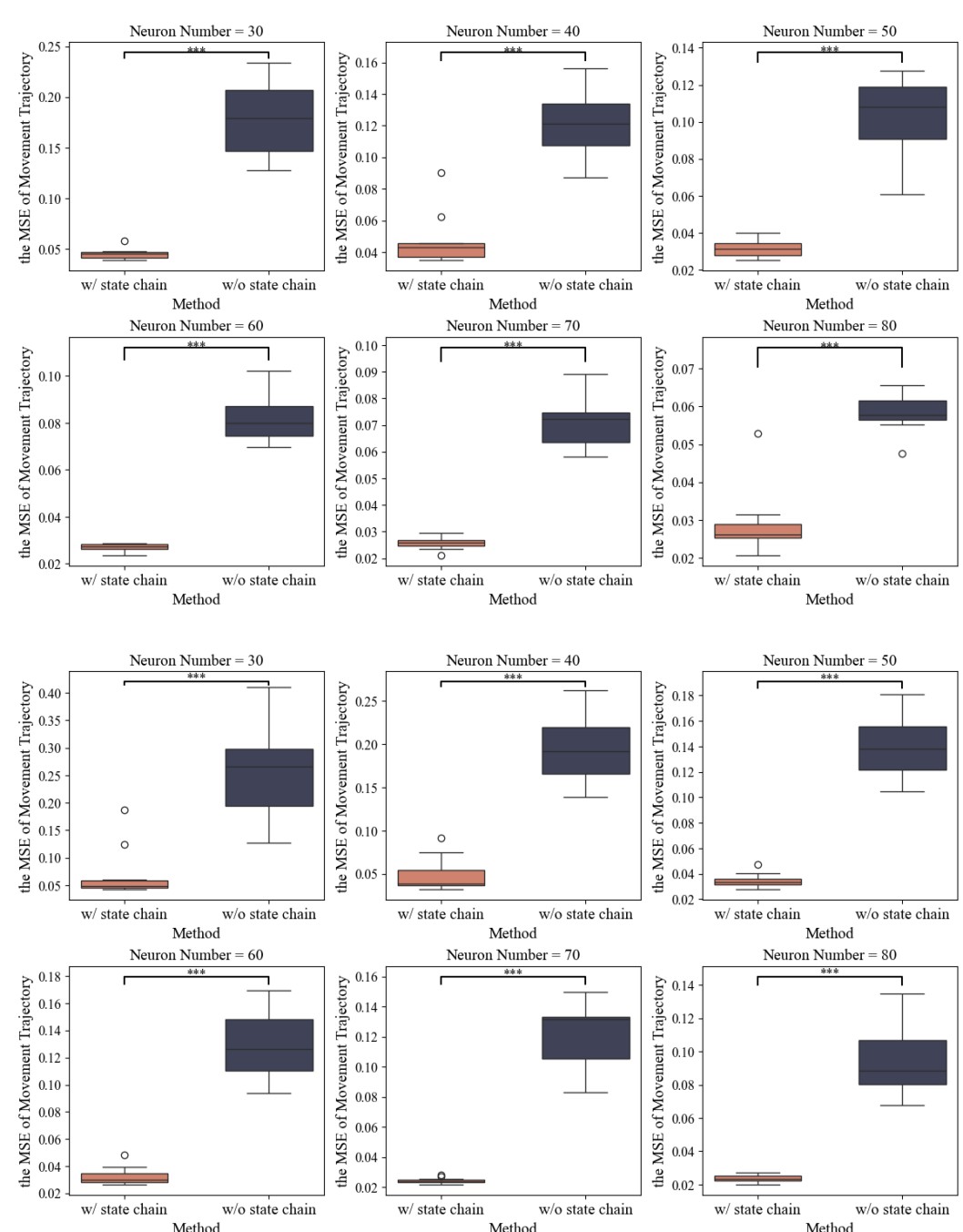

Figure S5: The statistical results of the MSE for the movement trajectory $x_{0:n}$ are shown in this graph. The number of neural states is set to 5 **(up)** and 10 **(down)**, and the number of neurons is set to $\{30, 40, 50, 60, 70, 80\}$. The decoding process is repeated 10 times with random initialization.

real-time decoding with approximately 20 ms per timestep. All experiments are completed in a Intel(R) Xeon(R) Platinum 8358 CPU @ 2.60GHz and a NVIDIA GeForce RTX 3090 GPU 24G.

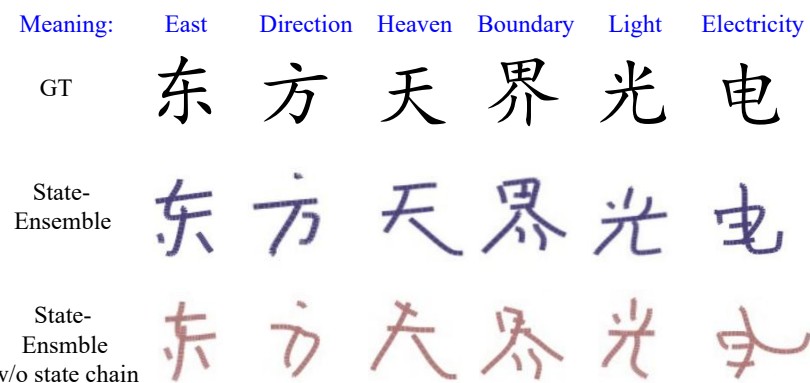

Figure S6: The characters decoded by the "with state chain" and "without state chain" methods are shown in this graph. Each column represents a different Chinese character, with the corresponding English translation displayed at the top.

## C.2 DETAILED SETTINGS OF DEEP LEARNING-BASED DECODERS

For the RNN, a single layer was employed to extract features, with the features from the final time step passed through a fully connected layer to generate the predictions. A grid search was conducted on one session to explore the following hyperparameters: the number of hidden units in $\{50, 100, 150, 200, \ldots, 500\}$, batch size in $\{32, 64, 96, 128\}$, dropout rate in $\{0, 0.1, 0.2, 0.3, 0.4, 0.5\}$, and learning rate in $\{5, 10, \ldots, 100\} \times 10^{-5}$. The optimal hyperparameters were selected based on the average performance on the validation set and applied across all sessions. The final configuration included 500 hidden units, a batch size of 32, a dropout rate of 0.3, and a learning rate of 0.001. For the LSTM, a similar procedure was followed. The chosen configuration consisted of 200 hidden units, a batch size of 32, a dropout rate of 0.3, and a learning rate of 0.001. For the Transformer-based decoders, we preserved their original architectures and configured them with a similar number of parameters as our model to ensure a fair comparison and to guarantee that the baseline models were adequately trained.

## C.3 DECODED TRAJECTORIES VISUALIZATION AND ANALYSIS

We visualize the estimated writing movement trajectories reconstructed using the "with state chain" and "without state chain" methods in fig. S6. The first row presents the ground truth (GT) trajectories of six example characters, while the second and third rows display the corresponding reconstructions using the "with state chain" and "without state chain" methods, respectively. The trajectories generated by the "with state chain" method exhibit more stable character structures and are noticeably closer to the GT. In contrast, the "without state chain" method results in significant distortions. For example, in the second character, the horizontal stroke is more accurately reconstructed, and in the sixth character, the vertical stroke with a hook appears more recognizable when using our method. This improvement can be attributed to the fact that neural state transitions are likely to occur when the writing speed changes within the same direction (e.g., a horizontal stroke) or when a stroke shifts direction (e.g., a vertical stroke followed by a hook) (Qi et al., 2025). By incorporating graph-based priors, the "with state chain" method can effectively capture these transitions, leading to more accurate and stable decoding outcomes.

# D DISCUSSION

## D.1 BROADER IMPACTS

This work introduces StateEnsemble, a dynamic model based on neural state chains. The model explicitly captures the neural state transition process, from which it extracts an embedded transition graph to enable state-dependent decoding. The transition graph derived from the state chain can reveal inherent patterns in human motion, thereby offering a robust prior for state-dependent neural decoding. Our experimental findings demonstrate that the incorporation of this prior knowledge leads to enhanced decoding stability and accuracy.

Furthermore, the proposed methodology holds promise for application in other complex motor tasks, such as cycling or piano playing. In such contexts, the extracted transition graphs could be used to identify patterns reflective of ingrained movement habits. These identified patterns, when subsequently employed as priors in state-dependent decoding, are anticipated to yield further improvements in the stability and precision of neural decoding across these diverse activities.

## D.2 LIMITATIONS

Current neuroscience research indicates that motion sequences are not encoded by a single neural state but rather by a sequence of distinct neural states, each characterized by a unique mapping between neural activity and kinematic parameters like velocity and position (Horrocks et al., 2024; Roth & Ding, 2024; Simpson, 2024; Qi et al., 2025). Despite this understanding, the collection of intracortical neural signals from the human motor cortex during the execution of complex movements remains a significant challenge. Therefore, the present study's evaluation of our proposed approach was necessarily limited to data from Chinese character writing, chosen as a representative complex motor task. Looking ahead, it will be crucial to extend this research by collecting intracortical neural data across a more diverse set of complex movements to thoroughly validate and generalize our findings.

## D.3 THE USE OF LARGE LANGUAGE MODELS

We employed large language models (LLMs) to assist in polishing and improving the clarity and readability of the manuscript.

| Method | Session1 | Session2 | Session3 | Session4 | Session5 | Session6 | Session7 |
|--------|----------|----------|----------|----------|----------|----------|----------|
| KF | 0.1434 | 0.0489 | 0.1192 | 0.1259 | 0 | 0 | 0.2875 |
| WF | 0.3364 | 0.2906 | 0.3129 | 0.2314 | 0.2330 | 0.1337 | 0.1398 |
| WCF | 0.3474 | 0.3143 | 0.3309 | 0.2408 | 0.2432 | 0.1349 | 0.1405 |
| RNN | 0.4090 | 0.3452 | 0.3486 | 0.3305 | 0.1815 | 0.2119 | 0.4966 |
| LSTM | 0.4079 | 0.3354 | 0.3437 | 0.3125 | 0.1840 | 0.2547 | 0.5419 |
| BIOT | 0.2715 | 0.2644 | 0.3322 | 0.2746 | 0.1258 | 0.1800 | 0.4280 |
| BrainBERT | 0.4075 | 0.3715 | 0.3764 | **0.3826** | **0.2589** | 0.2417 | 0.5052 |
| NDT2 | 0.4026 | 0.3782 | 0.3738 | 0.3659 | 0.2441 | **0.2672** | 0.5088 |
| Ours w/o | 0.4049 | **0.3786** | **0.3922** | 0.3731 | 0.2203 | 0.2638 | 0.5038 |
| Ours | **0.4151** | 0.3639 | 0.3640 | 0.3460 | 0.2357 | 0.2628 | **0.5441** |

| Method | Session8 | Session9 | Session10 | Session11 | Session12 | Session13 |
|--------|----------|----------|-----------|-----------|-----------|-----------|
| KF | 0.1964 | 0.0865 | 0.0918 | 0.1167 | 0.0591 | 0.2510 |
| WF | 0.1272 | 0.3249 | 0.3378 | 0.3304 | 0.1997 | 0.4338 |
| WCF | 0.1292 | 0.3344 | 0.3501 | 0.3384 | 0.2095 | 0.4454 |
| RNN | 0.3985 | 0.4354 | 0.4407 | 0.3826 | 0.2879 | 0.5120 |
| LSTM | 0.4135 | 0.4585 | 0.4695 | 0.4154 | 0.3307 | 0.5215 |
| BIOT | 0.3683 | 0.3335 | 0.3795 | 0.3432 | 0.2859 | 0.5131 |
| BrainBERT | 0.4574 | 0.3943 | 0.3888 | 0.4262 | 0.2850 | 0.5225 |
| NDT2 | 0.4470 | 0.4284 | 0.4168 | 0.3969 | 0.3235 | **0.5402** |
| Ours w/o | 0.4406 | 0.3543 | 0.4224 | 0.3724 | 0.3255 | 0.5093 |
| Ours | **0.4689** | **0.4637** | **0.4709** | **0.4275** | **0.3435** | 0.5401 |

| Method | Session14 | Session15 | Session16 | Session17 | Session18 | Average |
|--------|-----------|-----------|-----------|-----------|-----------|---------|
| KF | 0.2331 | 0.0406 | 0.0007 | 0 | 0.0139 | $0.1008 \pm 0.0893$ |
| WF | 0.4241 | 0.3159 | 0.2482 | 0.2196 | 0.0833 | $0.2624 \pm 0.0975$ |
| WCF | 0.4328 | 0.3258 | 0.2700 | 0.2359 | 0.0884 | $0.2729 \pm 0.1004$ |
| RNN | 0.4950 | 0.3173 | 0.2488 | 0.1902 | 0.2570 | $0.3493 \pm 0.1023$ |
| LSTM | 0.5419 | 0.3418 | 0.3108 | 0.2328 | 0.3023 | $0.3732 \pm 0.1020$ |
| BIOT | 0.4956 | 0.2494 | 0.1594 | 0.1438 | 0.1494 | $0.2943 \pm 0.1134$ |
| BrainBERT | 0.5201 | **0.3768** | 0.2847 | **0.2547** | **0.3131** | $0.3759 \pm 0.0871$ |
| NDT2 | 0.5167 | 0.3504 | 0.3024 | 0.2515 | 0.3016 | $0.3786 \pm 0.0863$ |
| Ours w/o | 0.5220 | 0.3244 | 0.2623 | 0.1957 | 0.2559 | $0.3623 \pm 0.0948$ |
| Ours | **0.5436** | 0.3434 | **0.3168** | 0.2416 | 0.3058 | $\mathbf{0.3887 \pm 0.0978}$ |

Table S1: $R^2$ values for all 18 sessions and the average between the true and estimated writing movement trajectories. The table compares the decoding performance of KF, WF, WCF, RNN, LSTM, BIOT, BrainBERT, NDT2, StateEnsemble without the state chain (Ours w/o SC), and StateEnsemble (Ours).

