# OpenReview forum: "StateEnsemble: Neural State Chain-based Dynamic Model for Complex Motor Sequence Decoding"
_ICLR.cc/2026/Conference — Submitted to ICLR 2026_

### Official Review · Reviewer_y2Bo · 2025-10-30

**Soundness:** 3
**Presentation:** 3
**Contribution:** 3
**Rating:** 6
**Confidence:** 2

**Summary:**

This paper presents a new approach to neural decoding under non-stationary conditions, based on a State-Ensemble Neural State-Space Model. The proposed framework extends standard state-space models by incorporating additional latent variables that capture slowly varying or non-stationary dynamics. The authors formulate a probabilistic ensemble of states, where each ensemble component evolves according to a learned transition model and contributes to the observation likelihood through a neural decoder. This structure allows the model to represent time-varying relationships between neural signals and latent behavioral states, while maintaining a coherent probabilistic interpretation. The paper validates the method on both synthetic datasets and real neural recordings, demonstrating improved robustness to non-stationarity and enhanced decoding performance compared to conventional Kalman-based and neural state-space baselines.

**Strengths:**

The paper presents a mathematically sound and principled method for neural decoding that explicitly models non-stationarity while building on well-established state-space model formulations. By integrating neural network components into a state-space architecture, the proposed framework effectively combines model-based structure with data-driven flexibility, providing a natural mechanism to adapt to distributional shifts in neural data.
The empirical section is well executed and supports the theoretical claims convincingly, with experiments conducted on both simulated and real neural datasets. The results consistently show improved decoding accuracy and stability over competing approaches. The presentation is clear, with strong mathematical rigor complemented by good conceptual explanations and experimental validation.

**Weaknesses:**

While the approach is well-formulated, the core idea of introducing additional state variables to handle non-stationarity is not entirely novel. Similar strategies have been explored in augmented-state Kalman filters and adaptive filtering literature, where time-varying parameters are embedded within an extended state representation. The paper would benefit from positioning its contribution more explicitly with respect to these prior works, clarifying what is truly new in the proposed state-ensemble formulation.
Additionally, the notation used in equation (5) is somewhat confusing: the augmented state transition is expressed using a function g(⋅) that is described as a conditional distribution function, whereas the remainder of the state-space model is formulated using deterministic mappings (e.g., f(⋅)). This notational inconsistency may obscure the conceptual distinction between stochastic modeling and deterministic dynamics, and should be clarified.

**Questions:**

In equation (13), is the relation presented an approximation (e.g., due to variational inference or an independence assumption), or should it in principle be an equality derived directly from the Markovian structure of the state-space model? Clarifying this point would help in understanding the scope and accuracy of the inference procedure.

---

### Official Review · Reviewer_xhh6 · 2025-10-30

**Soundness:** 3
**Presentation:** 2
**Contribution:** 3
**Rating:** 4
**Confidence:** 3

**Summary:**

This paper focuses on accurate movement decoding and introduces StateEnsemble, a neural state chain based model. It models the sequential relationships between neural states to perform state-dependent neural decoding. The authors analyze both simulated experiments and a handwriting task, demonstrating that the method outperforms baseline approaches in handwriting trajectory decoding.

**Strengths:**

The authors propose a novel method focusing on neural state decoding, with a clearly stated contribution. The experimental results on both simulated and handwriting data are promising and include comparisons with eight different baselines, demonstrating a thorough evaluation. The authors also clearly discuss the method’s limitations and potential impacts. The work is reproducible, with a detailed description of the algorithm, experimental setup, and baseline methods.

**Weaknesses:**

The main weaknesses of this work are organization and experiments.

The introduction lacks emphasis on the importance and applications of neural decoders. The background information and related work are scattered across different sections, which reduces the paper’s clarity and flow and makes it difficult to grasp the overall contributions of the method. Figure captions are not self-contained, and referencing appendix figures within the main text disrupts the flow.

Although the experiments on simulated data and neural signals are promising, the evaluation is limited, lacking comparisons with existing methods on larger, multi-participant datasets.

Please see the questions section for detailed suggestions related to both organizational and experimental weaknesses.

**Questions:**

**Organization and Presentation**

- The Introduction Section could be improved by focusing more on the importance of neural decoders, including examples of tasks and applications. This would help emphasize the motivation and significance of the study.

- The background information is spread across several sections, including Methods (the StataEnseble model section) and Experiments. It would improve the flow and clarity to organize the discussion of existing approaches and related work either within the Introduction or in a separate Background section. This could also improve the flow in the method section.

- Independent and descriptive figure captions would improve clarity. For example, including an explanation of the model framework directly in the caption of Figure 1 would strengthen the presentation.

- The presentation of appendix figures within the Experiments section should be changed. For example, Sections 3.2 and 3.3 refer to supplementary figures and explain them in the main text. These figures should either be moved entirely to the supplement with appropriate references in the main text, or incorporated into the main paper.

**Experiments**

- Does the handwriting dataset contain data from only a single participant? If yes, although the simulated and handwriting task results are promising, experiments using datasets with multiple participants are needed to better evaluate the method’s performance and contributions.

- Could the authors evaluate and compare their method with existing approaches on other public datasets that include multiple participants, such as those by Crell et al. [1] and Pei et al. [2]?
- What other tasks could this method be applied to?


Also, although the details on reproducibility are sufficient, it would be beneficial to the community if the authors could share the code in the future.

[1] Crell, Markus R., and Gernot R. Müller-Putz. "Handwritten character classification from EEG through continuous kinematic decoding." Computers in Biology and Medicine 182 (2024): 109132.
 [2] Pei, Leisi, Marieke Longcamp, Frederick Koon-Shing Leung, and Guang Ouyang. "Temporally resolved neural dynamics underlying handwriting." NeuroImage 244 (2021): 118578.

---

### Official Review · Reviewer_pNHD · 2025-10-31

**Soundness:** 2
**Presentation:** 2
**Contribution:** 1
**Rating:** 0
**Confidence:** 5

**Summary:**

This work describes a method of state-space based decoding using neural states and a transition graph applied a BCI handwriting task.

**Strengths:**

- Figure 1 and overall framework explanation was clear.
- It is a conceptually nice, clean framework: a series of states and a graph connecting them.

**Weaknesses:**

- Discussion of comparisons to other state-based approaches are lacking (e.g., Mamba, other dynamical modeling of latent neural dynamics using e.g. HMMs, switching LDSs, ...).  Qi 2019, 2022 is insufficient.
- The method description makes it hard to identify the novelty given how established neural state space models are. The use of DyEnsemble, and the whole section there doesn't appear novel (eqns 9-13).
- Figure 2 forces the reader to compare colors across matrices. Can a more direct metric be used as well? MSE is mentioned briefly but isn't in the main results.
- The use of Kendall tau distance assumes a lot "If the state transition graphs learned from these two groups are similar, it would suggest the presence of transition patterns in the neural state transition processes during handwriting tasks".  A more direct metric here would strengthen this analysis.
- Table 1 doesn't make clear which values are significantly different. Variances seem to heavily overlap.
- Results are very limited: limited comparison models, limited datasets as benchmarks (there are other animal and human handwriting and motor tasks available), limited metrics for evaluation, limited explanations for the performance (e.g., only compared with and without the state chain).

**Questions:**

- How well does the assumption ("We assume that the functional mapping between the neural signal and movement trajectory is stable within a state, while distinct among different states.") hold up in practice?  (Line 153).
- How well does this scale to longer chains? 10 appeared to be the maximum. Other work uses up to hundreds or thousands of states.

---

### Official Review · Reviewer_2Fiu · 2025-11-05

**Soundness:** 3
**Presentation:** 3
**Contribution:** 2
**Rating:** 4
**Confidence:** 3

**Summary:**

The paper introduces StateEnsemble, a novel neural decoding framework designed to improve brain–computer interface (BCI) performance, particularly for complex motor sequences such as handwriting. The key idea is to model motor control as a sequence of neural states (a “state chain”), with each state corresponding to a specific fragment of movement. Unlike traditional decoders or dynamic models that do not explicitly account for neural state transitions, StateEnsemble models the transition graph between neural states and uses it as a prior during decoding. The model is evaluated using (1) simulated data experiments, which show improved estimation of neural states and movement trajectories, and (2) human intracortical neural recordings from an imagined Chinese handwriting task. The results demonstrate higher decoding accuracy (R² = 0.3887) compared to Kalman Filter, LSTM, and Transformer baselines.

**Strengths:**

1.Introduces the idea of explicitly modeling neural state transitions as a graph prior—a promising direction for addressing neural nonstationarity in BCIs.

2.Includes both simulated and real intracortical data, lending credibility to the empirical findings.

3.The state transition graph provides interpretable insights into structured neural activity sequences, such as those involved in handwriting.

**Weaknesses:**

1.Although the approach is interesting, the scope of the work may be relatively narrow.

2.The method relies largely on existing components, and the theoretical contribution appears limited.

3.Most baseline comparison methods were published prior to 2024.

4.Despite the inclusion of a Reproducibility Statement, code and data availability are not clearly specified.

5.All real-data experiments are conducted on a single tetraplegic participant performing a specific handwriting task, leaving generalization to other subjects or tasks untested.

**Questions:**

1.Will you release the source code?

2.Can you include more recent SOTA baselines for comparison?

3.Can the method be evaluated on additional subjects or related tasks?

**Details Of Ethics Concerns:**

N.A.

---

### Meta-Review · Area_Chair_wp8T · 2025-12-22

**Summary:**

This submission proposes StateEnsemble, a neural decoding framework that explicitly models motor behavior as a chain of discrete neural states with a learned state-transition graph used as a prior during decoding. The method is evaluated on synthetic experiments and intracortical recordings from a handwriting task, reporting improved trajectory/decoding accuracy over classical filters and neural baselines, and offering some interpretability via the inferred transition structure. Reviewers generally agree the framework is clean and plausible and that the results are promising, but they raise substantial concerns about novelty/positioning vs. existing latent dynamical and switching/state-space approaches, as well as limited external validation (single participant/task), incomplete or dated baselines, and unclear code/data availability. These concerns drive a recommendation to reject at this time, while noting potential for a stronger resubmission with clearer novelty and broader evaluation.

**Reviewer Concerns:**

Authors did not post a rebuttal. All concerns are still outstanding.

**Reviewer Scores:**

Authors did not post a rebuttal. I expect the scores would have remained the same.

---

### Decision · Program_Chairs · 2026-01-26

Reject